

# Correction of estimation bias of predictive equations of energy expenditure based on wrist/waist-mounted accelerometers

Chin-Shan Ho[1,*], Chun-Hao Chang[1,*], Kuo-Chuan Lin[2], Chi-Chang Huang[1] and Yi-Ju Hsu[1]

[1] Graduate Institute of Sports Science, National Taiwan Sport University, Taoyuan, Taiwan
[2] Office of Physical Education, Chung Yuan Christian University, Taoyuan, Taiwan
[*] These authors contributed equally to this work.

Corresponding author
Chin-Shan Ho, kilmur33@gmail.com

## ABSTRACT

**Background**. Using wearable inertial sensors to accurately estimate energy expenditure (EE) during an athletic training process is important. Due to the characteristics of inertial sensors, however, the positions in which they are worn can produce signals of different natures. To understand and solve this issue, this study used the heart rate reserve (HRR) as a compensation factor to modify the traditional empirical equation of the accelerometer EE sensor and examine the possibility of improving the estimation of energy expenditure for sensors worn in different positions.

**Methods**. Indirect calorimetry was used as the criterion measure (CM) to measure the EE of 90 healthy adults on a treadmill (five speeds: 4.8, 6.4, 8.0, 9.7, and 11.3 km/h). The measurement was simultaneously performed with the ActiGraph GT9X-Link (placed on the wrist and waist) with the Polar H10 Heart Rate Monitor.

**Results**. At the same exercise intensity, the EE measurements of the GT9X on the wrist and waist had significant differences from those of the CM ($p < 0.05$). By using multiple regression analysis—utilizing values from vector magnitudes (VM), body weight (BW) and HRR parameters—accuracy of EE estimation was greatly improved compared to traditional equation. Modified models explained a greater proportion of variance ($R^2$) (wrist: 0.802; waist: 0.805) and demonstrated a good ICC (wrist: 0.863, waist: 0.889) compared to Freedson's VM3 Combination equation ($R^2$: wrist: 0.384, waist: 0.783; ICC: wrist: 0.073, waist: 0.868).

**Conclusions**. The EE estimation equation combining the VM of accelerometer measurements, BW and HRR greatly enhanced the accuracy of EE estimation based on data from accelerometers worn in different positions, particularly from those on the wrist.

# INTRODUCTION

Regular exercise is a cost-effective and efficient method of promoting and maintaining physical health and performance. Additionally, high levels of physical activity are closely
related to the decreased risk and mortality rates of chronic diseases (*Delisle et al., 2010*; *Kokkinos & Myers, 2010*; *Lawrence et al., 2014*; *González, Fuentes & Márquez, 2017*). In recent years, the accelerated development of smart products have made modern life more dependent on technology. This has resulted in a faster pace of life and significantly increased the pressures of work and life, leaving less time for exercise and rest. An increase in obesity rates across developing/developed countries has been associated with this technological era (*Zukiewicz Sobczak et al., 2014*; *Poobalan & Aucott, 2016*). Obesity has a detrimental impact on human health; it is now widely recognized that regular exercise and increased physical fitness is important to maintain one's health. Both the American Heart Association (AHA) and the American College of Sports Medicine (ACSM) have developed physical activity guidelines with recommendations for sufficient volumes and intensities of energy expenditure (EE). It is also recommended that treatment plans within the healthcare system include regular assessments of physical activity (*Garber et al., 2011*; *American College of Sports Medicine, 2019*; *American Heart Association, 2019*).

When assessing body weight management, inaccurate EE measurement can result in insufficient or excessive intake of nutrients, leading to a decreased amount of muscle tissues or the increased development of adipose tissues, respectively. This may also have significant implications for a person's health; the body's cardiovascular system, endocrine system, musculoskeletal system, thermoregulation, growth, and physical development can all be impacted (*Arieli & Constantini, 2012*; *Burrows et al., 2016*). Typical measurement methods of quantifying the EE and intensity of physical activity are doubly labelled water (DLW) and indirect calorimetry (*Gao et al., 2012*; *Brage et al., 2015*; *Beltrame et al., 2016*; *Lee et al., 2016*; *Westerterp, 2017*). Nevertheless, because of the costs and technical expertise required for these techniques, these methods are not commonly accessible in community settings. Due to continuous improvement of accelerometer-based wearable sensors, it is now easier to estimate EE and exercise intensity. As wearable sensors are easy-to-use and economical, and because they can continuously measure physical activity for a number of days (*Shih, Ho & Shiang, 2014*; *Montoye et al., 2015*; *Menai et al., 2017*; *Xu et al., 2018*), they have become a "popular" accessory for both general population and professional athletes who are undertaking exercise training.

The development of EE estimation equations has largely relied on treadmill exercise tests and daily activity tests as the experimental parameters, where the vector magnitude (VM) activity counts are used for the subsequent calculations (*Crouter, Clowers & Bassett Jr, 2006*; *Lyden et al., 2011*; *Sirichana et al., 2017*). The accelerometer-based wearable inertial sensors on the market are usually placed on the wrist or waist to provide referential EE estimates for different types of activities. Although some researchers reported that the wrist-mounted device can be used to estimate EE (*Welch et al., 2013*; *White et al., 2016*; *Sirichana et al., 2017*), recent research has demonstrated that waist-mounted devices worn closer to the body centre of mass can provide EE estimates that are more accurate than those of wrist-mounted devices (*Hildebrand et al., 2014*; *Tudor-Locke, Barreira & Schuna Jr, 2015*; *Chomistek et al., 2017*). It is also important to note that accelerometers can cause overestimation or underestimation of EE based on different exercise types or intensities

(i.e., cycling, uphill exercise, etc.) (*Schneller et al., 2015*; *Tarp, Andersen & Ostergaard, 2015*; *Yang et al., 2018*; *Kuo et al., 2018*; *Chang et al., 2019*).

The smart watch is currently a popular device worldwide and is widely discussed as a means to monitor EE. In recent years, the technology of combining the smart watch with the optical heart rate sensor has matured, improving HR (heart rate) monitoring. Due to the limitations of accelerometers, wearing the device on different parts of the body can produce acceleration signals of different natures and create bias in the data. Therefore, some researchers have used HR to modify customized exercise volumes in order to correct the measurement error caused by different wearing positions, to better estimate EE, and to obtain corrected parameters for modifying the EE estimation equation (*Domene & Easton, 2014*; *Oppert et al., 2016*; *Kuo et al., 2018*). Our previous study found that adding HRR parameters to the estimation equation could increase the accuracy of EE estimation more than HR parameters (*Chang et al., 2019*). Therefore, we hypothesised that the accuracy of EE estimation from wrist- and waist-worn accelerometers may be improved by applying a HRR correction. The purpose of this research was to modify the traditional EE estimation equation: Freedson VM3 Combination, 2011 (*ActiGraph, 2018*), which is suitable for devices worn on different parts of the body.

## MATERIALS & METHODS

### Study design

In this study, indirect calorimeter and accelerometer monitoring devices were used to measure the EE of healthy adults, and an accelerometer-based HR monitoring device was simultaneously used to measure their HR during the exercise tests. The HRR parameters were used to modify the EE estimation equation to make it applicable to both wrist- and waist-mounted devices. The study procedures were approved by the Institutional Review Board of Fu Jen Catholic University (Approval number: C106056). All participants were informed of the study procedures and provided written informed consent prior to the testing.

### Subjects

Ninety healthy adults voluntarily participated in this study. People who had exercise contraindications, were taking drugs that could change the metabolic rate, or had cardiovascular abnormalities were excluded from this research, as such circumstances could prevent them from completing the test safely. Subjects were asked to wear experimental devices and complete a one-hour test (including the time spent mounting the device on the body) in a laboratory at a room temperature of 23 °C (on average). If a subject failed to complete the test (e.g., failed to maintain a specified speed or withdrew before completion), the test was terminated immediately, and the subject's personal information and data were removed from the dataset. The subjects' anthropometric data are listed in Table 1.

### Measurement of EE by indirect calorimeter and accelerometer methods

Regarding the indirect calorimeter method, the Cardiopulmonary Exercise Testing System (Vmax Encore 29 System; VIASYS Healthcare Inc, Yorba Linda, CA) was adopted to
**Table 1 Anthropometric characteristics of participants.**

|  | Age (yrs) | Height (cm) | BMI (kg/m$^2$) | Body weight (kg) | Sex |
|---|---|---|---|---|---|
| Mean | 22.90 | 168.05 | 22.52 | 63.90 | 49 males, 41 females |
| SD | 4.15 | 7.62 | 3.25 | 12.06 | |

conduct the metabolic criterion measure (CM). The device was warmed up for at least 15 min and then calibrated before each of the tests. Test subjects wore a Hans-Rudolph mask that covered the nose and mouth. A sampling line and flow sensor connected to the mask were used to measure the volume and composition of inspired and expired air.

Regarding the accelerometer method, the ActiGraph GT9X-Link (ActiGraph Corporation, Pensacola, FL, USA) was used. This device measures accelerations across 3-axis and is small (3.5 cm × 3.5 cm × 1 cm) and lightweight (∼14 g). Before each test, the ActiLife6 software (version 6.12.1, Cary, NC, USA) was used to initialize the GT9X-Link and set the compatible chest-mounted Polar H10 Heart Rate Monitor (Polar Electro Oy, Finland). The sampling frequency of the monitoring device was 30 Hz, and the activity count and HR data were collected in 10-second epochs. The user manual of the ActiGraph recommends that the GT9X-Link was affixed to the wrist of the subject's non-dominant hand and the right hip on the midaxillary line (one on wrist, the other on waist) by an adjustable soft elastic belt. The test subjects were arranged to take one test at a time, and five treadmill speeds walking/running tests were conducted in a randomized order. During the process, VO$_2$ (oxygen uptake) by indirect calorimetry, HR and accelerometer counts for all tests were simultaneously and continuously recorded. The times of the Vmax and GT9X-Link were synchronously initiated according to the clock of the ActiGraph.

### Treadmill test

Subjects were requested to perform walking/running tests at speeds of 4.8, 6.4, 8.0, 9.7, and 11.3 km/h. Each speed test was at least 3 min, and all tests were separated by two-minute breaks (test method adapted from *Tudor-Locke, Barreira & Schuna, 2015*). If the subject failed to complete the test safely (e.g., was unable to maintain the treadmill speed) or the subject's heart rate exceeded the safety limit (220 minus the participant's age) during the exercise test, the test was terminated and any relevant data was deleted from the analysis.

### Data analysis

All ninety of the test subjects completed the exercise tests safely, therefore all Vmax, HR and GT9X-Link data were outputted into Microsoft Excel. The Vmax and HR data were used to calculate the parameters of the 10s-by-10s time series and were synchronized with the ActiGraph GT9X-Link accelerometer data. The data processing followed the approach of *Lyden et al. (2011)* whereby the first 120 s of each speed were removed to ensure that the participant achieves stability in movement under the exercise intensity. The VO$_2$ and carbon dioxide output (VCO$_2$) were calculated to determine EE by Weir's equation: EE (Kcals min$^{-1}$) = 3.491 (VO$_2$ in L/min) + 1.106 (VCO$_2$ in L/min) (*Weir, 1949*). The GT9X-Link data were analyzed by the GT9X-Link, and the EE was calculated based on

the *ActiGraph (2018)*. The equation is as follows: Kcals min$^{-1}$ = 0.001064 VM + 0.087512 BW − 5.500229. All of the EE values were divided by the weight for standardization and are presented in kcal min$^{-1}$ kgw$^{-1}$. The equation HRR = HR$_{max}$ − HR$_{rest}$ refers to the difference of HR$_{max}$ and HR$_{rest}$ for each phase of the test. HR$_{rest}$ is the pre-test measure of resting HR.

## Statistical analysis

The statistical software IBM SPSS Statistics version 20 (IBM Corp., New York, NY, USA) was used for statistical analysis. All data were summarized as means ± standard deviations. In order to understand how device placements affected EE estimation across different speeds, and the difference between criterion measured EE (CMEE) and device measured EE, one-way ANOVA with Games-Howell post hoc test, Cohen's d effect size (ES) and Mean Absolute Percentage Error (MAPE) were all calculated accordingly. Linear regression was used to modify the EE estimation model with the variables of VM activity counts, body weight and HRR. Validity and reliability of EE estimation was further evaluated using the criterion validity, namely the Pearson coefficient of determination (r) and Intraclass correlation coefficient (ICC), respectively. The significance level was set to $\alpha = 0.05$.

## RESULTS

The CMEE and GT9X EE (wrist and waist) data of the treadmill tests and the ES, MAPE and ICC of these two systems' measurement results are summarized in Table 2. The ANOVA test results of the CMEE and GT9X EE accelerometer data of the wrist- and waist-mounted devices indicated that in all speed test results, CMEE and GT9X EE data had significant differences ($p < 0.001$), and CMEE was higher than GT9X EE ($p < 0.001$, $t$-test with Games-Howell post hoc). The EE variation, estimated by the wrist-mounted device, was less than the CMEE and the waist-mounted device, resulting in a non-homogeneous condition. Therefore, the Game-Howell post hoc was used for verification. As shown in that Table 2, the GT9X EE (wrist and waist) values were lower than those of the CMEE, the wrist data had a high degree of difference in effect size (ES: 1.58 to 11.54) and MAPE (21.4 to 63.9%), and there was no significant correlation (r) in each individual speed as well as there being poor validity (ICC = 0.073). The waist data indicated a lower degree of difference in effect size (ES: 0.36 to 1.08) and MAPE (4.7 to 10.4%). A significant correlation was achieved at speeds 4.80, 6.42, and 9.66 km/h ($r = 0.529, 0.428$, and $0.210$, respectively) as well as good validity (ICC = 0.868).

Table 3 presents the results of multiple regression analysis of the EE estimation model composed of VM activity counts, body weight and HRR. The two modified models had high coefficients of determination ($R^2 = 0.802$ to $0.805$) and a small standard error of estimate (SEE). The criterion validity (r and ICC) results of CMEE and EE of different device positions estimated from the *ActiGraph (2018)* and modified models are listed in Table 4. From that table, it can be seen that the r and ICC values of the modified models (r: wrist = 0.895, waist = 0.897, very strong correlation; ICC: wrist = 0.863, waist = 0.889, good ICC) were greater than those of *ActiGraph (2018)* (r: wrist = 0.620, waist = 0.885, strong to very strong correlation; ICC: wrist = 0.073, waist = 0.868, poor to good ICC).

**Table 2  Comparison of measured EE by Vmax (CMEE) and estimated EE by GT9X-EE in 5 treadmill walking/running tests (mean ± SD).**

| Position | Treadmill Speed (km/h) | CMEE (kcal kgw$^{-1}$ min$^{-1}$) | GT9X-EE (kcal kgw$^{-1}$ min$^{-1}$) | ES | MAPE (%) | r | ICC |
|---|---|---|---|---|---|---|---|
| Wrist | 4.80 | 0.070 ± 0.009 | 0.055 ± 0.010[*] | 1.58 | 21.43 | 0.105 | |
| | 6.42 | 0.111 ± 0.014 | 0.065 ± 0.008[*] | 4.03 | 41.44 | 0.030 | |
| | 8.04 | 0.148 ± 0.010 | 0.071 ± 0.005[*] | 9.74 | 52.03 | −0.169 | 0.073 |
| | 9.66 | 0.172 ± 0.012 | 0.072 ± 0.005[*] | 10.88 | 58.14 | −0.165 | |
| | 11.28 | 0.202 ± 0.015 | 0.073 ± 0.005[*] | 11.54 | 63.86 | −0.130 | |
| Hip | 4.80 | 0.070 ± 0.009 | 0.066 ± 0.013 | 0.36 | 5.71 | 0.529[**] | |
| | 6.42 | 0.111 ± 0.014 | 0.104 ± 0.021[*] | 0.39 | 6.31 | 0.428[**] | |
| | 8.04 | 0.148 ± 0.010 | 0.141 ± 0.026[*] | 0.36 | 4.73 | 0.110 | 0.868 |
| | 9.66 | 0.172 ± 0.012 | 0.163 ± 0.023[*] | 0.49 | 5.23 | 0.210[**] | |
| | 11.28 | 0.202 ± 0.015 | 0.181 ± 0.023[*] | 1.08 | 10.40 | 0.162 | |

**Notes.**

[*]Significantly different from CMEE, $p < 0.05$.

[**]Significant correlation with CMEE, $p < 0.001$.

Mean values ± standard deviation (SD); CMEE, criterion measure energy expenditure; GT9X, ActiGraph GT9X-Link accelerometer; ES, Effect size (Cohen's d); Mean Absolute Percentage Error (MAPE) = {[ | (Predicted value - Actual value) |/Actual value] * 100}/n; r, Pearson coefficient of determination; ICC, intraclass correlation coefficient.

**Table 3  Modified models to predict EE (kcal kgw$^{-1}$min$^{-1}$) from VM, BW, and HRR.**

| Position | Prediction equation | $R^2$ | SEE |
|---|---|---|---|
| Wrist | EE = 0.000003 VM − 0.000461 BW + 0.000585 HRR + 0.078066 | .802 | 0.021 |
| Hip | EE = 0.000009 VM − 0.000299 BW + 0.000682 HRR + 0.046825 | .805 | 0.021 |

**Notes.**

VM, vector magnitudes; BW, body weight in kgw; HRR, heart rate reserve; $R^2$, coefficient of determination; SEE, standard error of estimate.

**Table 4  Validity and reliability analysis of traditional prediction equation and modified prediction equations.**

| Position | Freedson's VM3 Combination | | Modified models | |
|---|---|---|---|---|
| | r | ICC | r | ICC |
| Wrist | 0.620 | 0.073 | 0.895 | 0.863 |
| Hip | 0.885 | 0.868 | 0.897 | 0.889 |

**Notes.**

r, Pearson coefficient of determination; ICC, intraclass correlation coefficient.

## DISCUSSION

This study examined the effects of device measurement position on accelerometer outputs in estimating EE, and explored the possibility of modifying the estimation equation used previously. The main finding of this study was that the modified equation with HRR parameters significantly improved the accuracy of the estimated EE of the wrist-mounted device, while the waist-mounted device offers no significant improvement. From the results, it was clear that, when tested at different speeds, criterion measurement provided higher EE values than the wrist- and waist-mounted devices ($p < 0.05$), except for waist

EE at 4.8 km/h ($p > 0.05$). This study adopted *ActiGraph (2018)*, and the measurement results indicated that, although this equation underestimated the EE of devices worn on the waist, it still provided a good forecast (MAPE: 4.7 to 10.4%, ICC = 0.868). In addition, the EE estimated value bias of the wrist position was the greatest, and it was observed that as the speed increased, the GT9X EE prediction value of wrist did not increase significantly with speed and had poor prediction ability (MAPE: 21.4 to 63.9%, ICC = 0.073). If wrist-mounted monitoring is adopted, it will affect not only EE estimates, but it potentially could also affect health and/or fitness outcomes for the user. According to our previous study, HRR parameters can be used to calibrate differences in physical fitness and standardize an individual's physical fitness level (*Chang et al., 2019*). In this study, HRR parameters were also added to the traditional estimation equation (*ActiGraph, 2018*). This addition indeed improved the validity of the EE estimates of wrist- and waist-mounted devices. The ICC of modified models in the wrist position was 0.863, and that of the waist position was 0.889. This strongly suggests that the combination of HRR, accelerometer outputs and body weight can increase the accuracy of EE estimates for devices worn on different parts of the body.

Accelerometer-based EE sensors carry out calculations using VM activity counts produced during physical activity. A limitation of the accelerometers in the sensors is that they can create different vibration signals due to differences in the positions in which the devices are worn. In other words, with the same exercise intensity, VM activity counts can increase or decrease due to differences in device positions, leading to bias in EE estimates. In the current study, it was found that the speed measurement results of the wrist position were underestimated ($p < 0.05$), that there was a large ES (wrist: 1.58 to 11.54), and that the bias could increase following increases in speed. The accelerometer-based EE estimation models were built on a database of healthy adults in which the subjects wore accelerometers on the waist for exercise tests (*Crouter, Clowers & Bassett Jr, 2006*; *Lyden et al., 2011*), therefore, such models are more suitable for estimating the EE measured from the waist-mounted devices. Despite a significant difference between the GT9X EE data of the waist-mounted device and the criterion measurement in all speed tests ($p < 0.05$), it still provided a lower degree of difference in the ES (waist: 0.36 to 1.08), good validity ($r = 0.885$) and reliability (ICC = 0.868). Therefore, when ignoring the difference in vibration signals produced by devices worn in different positions, this kind of non-standard measurement methods (i.e., physical activity sensors) may create different levels of bias.

As the waist is located closer to the center of body mass than wrist, it is the ideal position for wearing an accelerometer device for measuring a subject's overall physical activity. Recent researchers have also confirmed that compared to wrist-mounted accelerometer-based devices, waist-mounted devices can provide more valid (and/or reliable) estimates of EE (*Hildebrand et al., 2014*; *Tudor-Locke, Barreira & Schuna Jr, 2015*; *Chomistek et al., 2017*). Nevertheless, smart watches and wrist-mounted sensors have become a trend in recent years. Therefore, it is necessary to increase the accuracy of wrist-mounted EE monitoring devices. Furthermore, it is important to note that EE (metabolism) is a complicated physiological mechanism that cannot be accurately measured using only the accelerometer. In addition, physical activity may be misclassified if accelerometers are

placed in different positions which then produce signals of different strengths. Based on the results of the current investigation, when the EE estimation equation is modified it is recommended to take physiological parameters into consideration. All physical activity involves muscle contraction, which has an energy cost. When the skeletal muscle is moving, the autonomic nervous system will regulate the functions of the cardiovascular system to satisfy the energy and metabolism demands. Moreover, the control of the autonomic nervous system over HR is a dynamic regulation process (*Fisher, Young & Fadel, 2015*; *Dong, 2016*; *Chen et al., 2017*). Much previous research has also indicated a close linear correlation between HR and $VO_2$ which is why changes in HR can be used to evaluate exercise intensity (*Strath et al., 2001*; *Bouchard & Trudeau, 2008*; *Villars et al., 2012*; *Domene & Easton, 2014*; *Rousset et al., 2015*; *Colosio et al., 2018*).

As a result, many researchers have suggested that accelerometer-based EE sensors can be used in conjunction with HR monitoring devices to increase the accuracy of EE estimates. *Domene & Easton (2014)* indicated that Latin dance has relatively more gestures than other forms and discussed the validity of combining wrist- and waist-mounted triaxial accelerometers with heart rate measuring devices, and they evaluated the physiological and activity parameters of Latin gestures. In their study, they recruited 22 dancers to estimate their EE using accelerometer and HR data, and they modified the estimation equation based on the EEs measured using indirect calorimetry. Their results determined that wrist-mounted accelerometers combined with HR data provide accuracy comparable to criterion measurement (indirect calorimetry) results. *Kuo et al. (2018)* discussed the possibility of improving the EE of uphill hiking using an EE estimation model established based on accelerometer and HR parameters. Their results indicated that the estimation equation could also increase the accuracy of estimating the EE of uphill hiking when the parameters of the accelerometer and HR changes were included. *Chang et al. (2019)* also studied the accuracy of EE estimates in an uphill walking or running activity using accelerometers, where HR and HRR were used to modify the traditional empirical equation of accelerometers as compensation factors. They found that, compared with HR, HRR provides better estimates when it is combined with the parameters of the accelerometer. The results of this study are consistent with those of previous research. As it has been proven that the initial value of HR can be variable due to an individual's physical fitness or psychological factors and can affect the accuracy of EE estimation, HRR can therefore be used as a compensation factor to correct for the user's physical fitness and improve the estimate of exercise intensity, making the estimate closer to the actual value. The mean error rates of GT9X-EE and CMEE derived from the modified models were 8.0% for the wrist and 5.2% for the waist. The difference shows that the EE estimate of wrist-mounted devices greatly reduced the mean error rate (from 47.4% to 8.0%) and increased the validity (r: 0.620 to 0.895), and reliability (ICC: 0.073 to 0.863).

It is important to accurately measure or estimate EE during physical activity or exercise. A key factor in enhancing training outcomes is ensuring individuals have an adequate energy balance, that is, energy intake and EE are matched. Inaccurate EE estimates can result in insufficient or excessive intake of nutrients, leading to decreases in muscle tissues or increases in adipose tissues, respectively. Such changes can not only affect an individual's

athletic performance but also have a negative impact on their health. Moreover, HRR, which is the difference between maximal heart rate and resting heart rate, can correct for excessive resting heart rates caused by differences in physical fitness and be used for estimating EE or exercise intensity. This study and our previous study strongly indicate that the prediction equation contains HRR parameters, the main difference between *ActiGraph (2018)* (traditional EE prediction equation) and the modified model is the HRR factor, which has better EE prediction ability than the traditional EE prediction equation. Including HRR not only improved the reliability and effectiveness of estimates but also increased the accuracy of the estimation model, particularly that of devices worn on the wrist.

## CONCLUSIONS

In conclusion, the *ActiGraph (2018)* is unlikely to accurately measure the EE by using wrist-mounted accelerometer-based physical activity sensors, for the values can be somewhat underestimated. This study reports that the vector magnitude parameters of accelerometer, body weight and HRR parameters can improve EE estimates, particularly those based on data from wrist-mounted accelerometer-based physical activity sensors. Although most of the commercial smart watches today have built-in accelerometer-based physical activity monitoring modules and optical heart rate sensors on the bottom, it is interesting to note that the data of these two devices are not integrated. Therefore, we suggest that the data of these two devices and our study results can be combined to develop a more accurate and reliable form of EE measurement technology.

### Funding
This study was supported by the Ministry of Science and Technology, Taiwan, under Grant MOST 107-2410-H-179-007. The funders had no role in study design, data collection and analysis, decision to publish, or preparation of the manuscript.

### Grant Disclosures
The following grant information was disclosed by the authors:
Ministry of Science and Technology, Taiwan: MOST 107-2410-H-179-007.

### Competing Interests
The authors declare there are no competing interests.

### Author Contributions
- Chin-Shan Ho conceived and designed the experiments, contributed reagents/materials/analysis tools, authored or reviewed drafts of the paper, approved the final draft.
- Chun-Hao Chang conceived and designed the experiments, performed the experiments, analyzed the data, prepared figures and/or tables, authored or reviewed drafts of the paper, approved the final draft.

- Kuo-Chuan Lin conceived and designed the experiments, analyzed the data, authored or reviewed drafts of the paper, approved the final draft.
- Chi-Chang Huang performed the experiments, contributed reagents/materials/analysis tools, authored or reviewed drafts of the paper, approved the final draft.
- Yi-Ju Hsu performed the experiments, prepared figures and/or tables, authored or reviewed drafts of the paper, approved the final draft.

### Human Ethics

The following information was supplied relating to ethical approvals (i.e., approving body and any reference numbers):

The Institutional Review Board of Fu Jen Catholic University granted Ethical approval to carry out the study within its facilities (registration number: C106056).

### Ethics

The following information was supplied relating to ethical approvals (i.e., approving body and any reference numbers):

The Institutional Review Board of Fu Jen Catholic University granted Ethical approval to carry out the study within its facilities (registration number: C106056).

### Data Availability

Raw data is available as a Supplemental File.

### Supplemental Information

Supplemental information for this article can be found online at http://dx.doi.org/10.7717/peerj.7973#supplemental-information.

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
