# Peer review of "Correction of estimation bias of predictive equations of energy expenditure based on wrist/waist-mounted accelerometers"

_PeerJ, doi:10.7717/peerj.7973_

## Round 0.1 · original submission · Major Revisions

Thank you for considering PeerJ for your research. As you will see from the reviewer comments both agree the article has the potential to add to the literature. However, both also identified that the writing style needs significant improvement to assist with the readability of the paper, in addition to other comments related to the scientific content of the paper. In order to be suitable for PeerJ you will need to significantly revise the paper and show evidence of a thorough review of the writing style (which may require you to get professional input). I hope you do take the time to revise your paper and I look forward to seeing a revised version.

Reviewer 1 ·

Basic reporting

Overall the English language needs great improvement to ensure an understanding from a wide audience. Below is an extensive list of suggestions.

Abstract

- Line 31 – Last sentence needs clarity. Specifically, need definition for EHG (or just say ‘hip’?).
o May be improved by stating that by using multiple regression using VM’s, BW and HRR, accuracy of energy expenditure estimation was greatly improved compared to traditional formula (provide the R2 and ICC of original and modified models)

Introduction

- Line 41 to 44 – revise this. Consider words to the effect of “Regular exercise is a cost-effective and efficient method of promoting and maintaining physical health and performance. Additionally, high levels of physical activity are closely related to decreasing the risk and mortality rates of chronic diseases.”
- Line 45 to 48 – used the word ‘technology’ twice and the sentence is difficult to read. Sentence that starts “making obesity..” does not make sense.
- Line 48 – revise this. Consider words to the effect of “Obesity has a detrimental impact on human health, it is now widely recognised that regular exercise and increased physical fitness is important…”
- Line 51 to 52 – Change ‘amounts’ to ‘volumes’. Remove the full stop after ‘intensities’ and decapitalise the ‘To’.
- Line 52 – revise this. Consider words to the effect of “it is recommended that treatment plans include regular assessments of physical activity as part of the healthcare system”
- Line 57 – after ‘adipose tissues’, add ‘, respectively’, this indicates to the reader the respective implication of insufficient or excessive intake of nutrients.
- Line 57 – revise this. Consider words to the effect of “This may also have significant implication for one’s cardiovascular system, endocrine system……
- Line 59 – revise this. “The gold standard methods of quantifying the EE and intensity and of physical activity are DLW and indirect calorimetry”
- Line 62 to 63 – revise. “nevertheless, these methods are not always practical or feasible due to the site requirements and installation costs”
- Line 67. Change “must-have” to “popular”.
- Line 69. Change to “the development of EE estimation formulas has relied…”
- Line 75 - revise this. Consider words to the effect of “due to its convenience and ability to provide good EE estimation values”
- Line 77 – change to “..devices worn closer to the bodies centre of mass…”
- Line 79 – can you provide more information? What activities have been shown to be less accurate (i.e. resistance exercise? Cycling? Etc..)
- Line 84. Define ‘HR’ on first use.
- Line 87. Change to “Therefore, some researchers have used HR to modify…”
- Line 91 – provide the reference for you previous study
- Line 91 to 94 – revise. Consider words to the effect of “Due to this, we hypothesise that accuracy of EE estimation from wrist- and waist-worn accelerometers will be improved by applying a HRR correction.
- Line 84 – Used the word ‘formula’ twice, consider changing.
- Line 96 – You have previously stated that wrist accelerometers can provide good estimates of EE (line 75), and in this line you are stating that they cannot estimate EE accurately. Consider revising this and making the narrative consistent.

Materials and methods

Study Design

- Line 101. Make clearer what methods were used for EE measurement / estimation and what was used for HR monitoring.
- Line 102. What is ‘general adults’? Perhaps changes to ‘healthy adults’.

Subjects

- Line 112. Remove ‘These’ and just begin the sentence with ‘Subjects’.
- Line 115. Change ‘gave up before finishing’ to ‘withdrew before completion’
- Table 1
o Change ‘weight’ to ‘body mass’
Measurement of EE…..
- Line 120. Change ‘In the..’ to ‘Regarding the…’
- Line 126 to 127. Revise this as does not flow well. Consider words to the effect of ‘ Regarding the accelerometer method, the ActiGraph GT9X-Link (ActiGraph Corporation, Pensacola, FL, USA) was used. This device measures accelerations across 3-axis and is small (3.5 cm x 3.5 cm x 1 cm) and lightweight (~14 g)’
- Line 130. Remove ‘in this study’ and begin sentence with ‘The sampling frequency…’
- Line 136. Define VO2.
- Line 137. You have said that ‘counts for all tests will be simultaneously…’ This sentence is in a different tense. Suggest sticking to the past tense and change to ‘counts for all tests were simultaneously….’

Treadmill Test

- Line 142. Remove ‘In the laboratory’ and begin sentence with ‘Subjects were…’
- Line 145. Is maximum HR your safety range, if so why? Did you have any other safety considerations?

Data Analysis

- Line 151. Change ‘and all Vmax…’ to ‘therefore all Vmax…’. Avoids the repetition of the word ‘and’
- Line 152. Include ‘Microsoft Excel’ or ‘MS Excel’
- Line 154. Revise this. Consider words to the effect of ‘The data processing followed the approach of Lynden et al. (2011) whereby the first 120 seconds and last 10 seconds of each speed were removed to ensure the participants were in a steady state’
- Line 158. Define VCO2. before abbreviate

Statistical Analysis

- Line 166. Revise this. Consider words to the effect of ‘In order to understand how device placements affected EE estimation across different speeds, and the difference between criterion measured EE (CMEE) and device measured EE, one-way ANOVA with Games-Howell post hoc test, Cohen’s d effect size (ES) and Mean Absolute Percentage Error (MAPE) were calculated accordingly.’
- Line 170 – Remove ‘In this study,’ and begin sentence with ‘Linear regression was used…)
- Line 171 - Revise this. Consider words to the effect of ‘Pearson’s coefficient of determination and Intraclass correlation coefficients (ICC) were used to further evaluate the relationship and reliability of EE estimates, respectively.
- Line 173. Move the sentence starting ‘The statistical software IBM…’ to the beginning of the statistical analysis section.

Results

- Line 181. Change ‘(p < 0.000)’ to ‘(p < 0.001)’
o And do this throughout (e.g Line 182)
- Line 190. Remove ‘(R2) as it is repeated in the next bracket.
- Line 193. Change ‘bigger’ to ‘greater’
- Line 195. You have stated that this is a ‘high ICC’. Have you got a reference for this assessment of magnitude?
o See the below reference for reporting guidelines for ICC
o https://www.sciencedirect.com/science/article/abs/pii/S1556370716000158
o Allows ease of interpretation. You can then say in the discussion that the ICC went from ‘moderate’ to ‘good’ when correcting for HRR.
- Line 195 to 199. This may be better suited for the discussion. Results sections should just state the results.

Discussion

- Line 203 to 204 – change ‘in the past’ to previously’
- Line 204 to 206. Change this sentence to make it clearer what the key results were.
- Line 213. Have you a reference for your previous study? If so, put it in there.
- Line 227. You shouldn’t being a sentence with ‘Because’
- Line 255. At the end of the sentence it says ‘E03r”, what is that?

Experimental design

Experimental design

Methods

Aside from some changes required to the grammar and flow, generally I think the method section is good, easy to follow and is designed appropriately to assess your research question. Likewise with the statistical analysis, it is easy to follow and easy to replicate.
You have stated that you have used ICC to assess the reliability of measurements. What is your justification for this? An assessment of reliability would usually be appropriate when administering the same test twice on the same individuals, and assessing the closeness in results from each device. In this study, is it not the validity that you are assessing?

- Line 162. Regarding your HRR calculation, you have said that HRR = HRmax - HRrest and they refer to the difference for each phase of the test. To clarify, is the HRrest measurement the lowest recorded HR taken in the 3 minute break in between speeds? Or a pre-test measure of resting HR? Just needs a little bit more clarity.

Validity of the findings

Statistical analysis

- Table 2

o Would be better suited to provide Pearson’s correlation in this table?
 Could be worth providing the correlations for each individual speed? To provide a greater depth of information
o As it stands it looks as if the MAPE is just for the 8.04 km/h speed as it is aligned with it
 As above, think it would be good to show the MAPE for each individual speed.

- Games-Howell post hoc test was used. Does this mean your data were not normally distributed? If so, it may be worth saying that and stating briefly how you assessed the distribution.

- Thank you for providing your data. Given the clarity in your methods section I was able to run the same analyses on your data set and came up with identical results for all but one test.
o You have reported in table 4 that the r and ICC for the hip estimated EE Freedson model to be 0.624 and 0.612, respectively. In doing the same analysis I got an r and ICC of 0.885 and 0.868, respectively. Could you double check this?
 This may have implication for your discussion as this would mean that the addition of HRR to the equation for hip estimated EE only improves the model in a minor way (r and ICC of modified model is 0.897 and 0.889, respectively).

Additional comments

Thank you for the opportunity to review this work. Although I think the article needs significant work regarding how it is written and structured, of which I have gone through and made specific comments on, I believe the data is valuable and demonstrates the importance of collecting further physiological information in order to improve the accuracy of energy expenditure estimation from accelerometers. Generally I think the statistical analysis is good and methods easy to follow. I have included some comments on some minor changes which may improve the manuscript for publication. Furthermore, when analysing your data following the same process as you have stated I found slightly different results regarding the hip worn estimates, I have made further comments on this below.

Reviewer 2 ·

Basic reporting

Introduction
Para 1, line 41: ‘Doing regular and correct exercise…’ – what is correct exercise? Do you mean perhaps mean appropriately prescribed exercise or similar?
Para 1, line 41: ‘Doing regular and correct exercise is one of the cheaper and efficient way…’ Cheaper compared to what? Perhaps modify to ….’a cheap and efficient way….
Para 1, line 42: ‘Not only have its benefits…’ – suggest you modify to ‘Not only has exercise’s benefits…
Para 1, line 47: ‘Making obesity is one…..’ – suggest you modify to ‘Associated with this has been an increase in obesity rates across westernised countries (insert reference)’ or similar.
Para 1, line 51-52: “To increase….exercise’ – suggest deleting this sentence.
Para 2, line 55: “From the aspect of weight management’ – suggest you change to ‘From the aspect of body weight management’
Para 2, line 60: Suggest replace ‘metabolic cart’ with ‘indirect calorimetry’. Also, a heat-exchange chamber (direct calorimetry) is actually the ‘gold standard’ for measuring EE.
Para 2, line 60: ….can quantify the intensity and EE of all types…. – suggest you modify to …..can quantify the rate and total EE…..
Para 2, line 62: ‘Nevertheless, because of their….can afford them.’ – suggest you modify to ‘ ..because of the costs and technical expertise required for these techniques these methods are not commonly accessible (in community settings?)’ or similar
Para 2, line 63: Suggest replacing ‘optimization’ with ‘improvement’.
Para 2, line 68: Who is a ‘general user’? Do you mean general population?
Para 2, line 68: Suggest replacing ‘conducting’ with ‘undertaking’
Para 4, line 92: ‘we hypothesis’ – change to ‘we hypothesised’
Para 4, line 94: Modify what EE estimation formula? Please be specific.
Materials & Methods
Line 102: Suggest replacing ‘adopted’ with ‘used’
Line 102: What is a general adult?
Line 115: ‘…specified speed or gave up before finishing)’ – do you mean ‘terminated prior to completion of the test’?
Line 124: Suggest changing ‘amount or air….of O2 and CO2.’ to ‘volume and composition of inspired and expired air’ or similar.
Line 127: Suggest delete ‘activities’
Line 137: Replace ‘will be’ with ‘were’
Line 171: “weight’ – do you mean body weight?
Results
Line 189: as per above.
Line 190: What is a ‘rather small’ SE – Try to avoid subjective descriptions. Was is small, trivial, moderate, etc., etc.?
Line 193: …were ‘bigger’ than those – suggest replacing with ‘larger’
Lines 196-199: ‘From the aforesaid results…….worn on the wrist’ – suggest condensing these two sentences to ‘Including HRR not only improved the reliability and…….worn on the wrist’.
Discussion
Line 205: ‘…provided higher EE estimate than…. - the CMEE is not an estimate, it was measured via indirect calorimetry?
Line 209: ‘Bias’ of the wrist position? Do you mean error associated with the wrist position?
Line 211-213: “if the approach ….accuracy of EE’ – suggest simplifying this message to ‘If wrist-mounted monitoring is adopted, it will affect not only EE estimates but potentially also health and/or fitness outcomes for the user’ or similar.
Lines 213-214: Insert reference.
Line 215: ‘were also added to the estimation formula’ – what formula? Please be specific with this detail.
Line 221: Suggest replacing exercise with activity.
Line 223: Suggest deleting ‘obviously’
Line 224: Suggesting changing ‘In our study’ to “In the current study’
Line 225: Suggest deleting ‘obviously’
Line 226: Suggest changing ‘there was a big difference in ES..’ to ‘there was a large ES…’
Line 227-233: I am not entirely clear on the message(s) in these two sentences. Perhaps it is just me but I feel that the clarity can be improved.
Line 234: What is a ‘non-indirect calorimeter approach’ – do you mean an alternative to indirect calorimetry?
Line 236: ‘As the waist is the location……’ – compared to what? Closer to the COM compared to wrist-mounted devices?
Line 238: Edit ‘Researches’ – do you mean ‘researchers’?
Line 239: ‘….than are more close to reality’ – do you simply mean waist-mounted devices can provide more valid (and/or reliable) estimates of EE?
Line 244: Suggest replacing ‘misjudged’ with ‘misclassified’
Line 245-247: ‘Therefore, when the EE……into consideration.’ – Is it absolutely necessary? Is there evidence to indicate this is always necessary? What is movement detection technology improves? Or are you suggesting based on the results of the current investigation that the addition of physiological data can improve accelerometer-derived estimates of EE?
Lines 247-248: Suggest changing ‘All physical activities are driven by antagonism of skeletal muscle…’ to ‘All physical activity involves muscle contraction, which has an energy cost.’ or similar.
Line 255: What is E03r? Suggest this should be deleted?
Line261: Suggest replacing ‘solicited’ with ‘recruited’
Line 263: Suggest replacing ’metabolic carts’ with ‘indirect calorimetry’
Line 264: What do you mean by ‘provide reliability as high as the standard measurement results’? what is the standard measurement?
Line 276: What do you mean by’ …make the estimate close to the actual value’ – do you mean improve the estimate of exercise intensity?
Line 280: Suggest changing ‘…..exercise or training process’ to ‘physical activity or exercise’.
Lines 281-282: Suggest modifying this sentence ‘To maintain….are identical’ – it reads as though there are two themes being addressed. You could consider deleting ‘To maintain the energy balance and monitor the training amount’, and Begin the sentence with ‘A key factor in enhancing training outcomes is ensuring …’
Line 284: ‘issues’ do you mean tissue?
Line 284: Insert “an’ after ‘affect’
Lines 285-286: ‘…but also create……and growth retardation’ - these are two more extreme examples of inadequate energy intake, perhaps you could simply say ‘but also have a negative impact on health’ or similar.
Lines 288-290: HRR can better estimate EE compare to what?
Conclusions
Line 293: What ‘current EE estimation formula’?
Line 296: ….’better modify’…. – do you mean improve?
Line 301-302: Suggest changing ‘…..and even more convincing….’ to a more objective statement (e.g. more accurate and reliable).

Experimental design

The study appears to be well controlled, with a good sample size. However, it is questionable as to whether participants reached a 'steady state' after 2 min for subsequent expired air and HR analysis over the final 60 s of each speed.

Validity of the findings

No comment

Additional comments

Wearables devices, and their subsequent application to health and physical fitness monitoring in the general population is becoming increasingly popular. Despite this popularity, there is a relative dearth of empirical data to support many of these devices or the application of the measures (e.g. EE prediction from accelerometry). This manuscript can play a role in improving our understanding of the validity (and limitations) of current device types (i.e. wrist-mounted accelerometers), and potentially help to inform future improvements. Some improvements in the language and clarity of the message would greatly improve the manuscript, which would no doubt be of interest to many readers.

---

## Round 0.2 · Minor Revisions

Thank you again for taking the time to revise your paper. I think it has now been greatly improved. There remain some minor edits though that are required before it can be finalised. Please see the reviewer comments for more detail.

Reviewer 1 ·

Basic reporting

Basic reporting
The grammar and language used in this revised manuscript is much improved on the previous version. Below is some minor suggestions.
Abstract
Line 25. Capitalise ‘Indirect’ at the beginning of the methods section. Change ‘calorimeter’ to ‘calorimetry’
Line 33. Revise this as sounds a little confusing. Consider words to the effect of. ‘Modified models explained a greater proportion of variance (R2: wrist = 0.802; waist = 0.805) and demonstrated a good ICC …..).
Line 37. Remove the comma after HRR
Line 38. Remove ‘and’
Introduction
Line 61. Consider this. ‘Typical measurement methods of quantifying the EE and intensity of physical activity are doubly labelled water (DLW) and indirect calorimetry’
Line 66. Consider this. ‘… it is now easier to estimate EE and exercise intensity’
Line 67. Try to avoid starting sentences with ‘Because’. Change to ‘As’
Line 71. Consider this. ‘.. EE estimation equations has largely relied on..’
Line 84. Remove the comma after worldwide.
Line 88. Change ‘Because of this’ to ‘Therefore, some researchers….’
Materials and methods
Line 106. Remove ‘, and’ and insert a full stop ‘…C106056). All participants…’
Line 173. Remove ‘Then the’ and begin sentence with ‘Validity’
Results
No changes required
Discussion
Line 206/207. Consider this. ‘…HRR parameters significantly improved the….’
Line 209. Remove ‘did’
Line 287. Consider revising this, along the lines of. ‘A key factor in enhancing training outcomes is ensuring individuals have an adequate energy balance, that is, energy intake and EE are matched’
- Although I don’t think you mean that, energy intake and expenditure do not need to be ‘identical’ to enhance the training process.

Experimental design

Satisfied with the revisions and explanation

Validity of the findings

Satisfied with the revisions and explanation

Additional comments

Thank you for the opportunity to review this version of the manuscript. This article is much improved and the writing style and structure will enable an easier interpretation for readers. As mentioned previously, I believe this is a good paper which employs suitable methods to answer the research question. I have included some very minor changes regarding the use of language/grammar.

---

## Round 0.3 · accepted · Accept

Thank you very much for taking the time to revise the paper. I think you will agree it is now a better article as a result. Congratulations on having your paper accepted.